# Deconvolving Complex Neuronal Networks into Interpretable Task-Specific Connectomes

## Abstract

Neuronal responses associated with complex tasks are superpositions of several elementary physiological and functional motifs. Important challenges in this context relate to identification of elementary responses (also known as basic functional neuronal networks), combinations of responses for given tasks, and their use in task and efficacy prediction, and physiological characterization. Task-specific functional MRI (fMRI) images provide excellent datasets for studying the neuronal basis of cognitive processes. In this work, we focus on the problem of deconvolving task-specific aggregate neuronal networks into elementary networks, to use these networks for functional characterization, and to "explain" these networks by mapping them to underlying physiological regions of the brain. This task poses a number of challenges due to very high dimensionality, small number of samples, acquisition variability, and noise. We propose a deconvolution method based on supervised non-negative matrix factorization (SupNMF) that identifies elementary networks as factors of a suitably constructed matrix. We show the following important results: (i) SupNMF reveals cognitive "building blocks" of task connectomes that are physiologically interpretable; (ii) SupNMF factors can be used to predict tasks with high accuracy; and (iii) SupNMF outperforms other supervised factoring techniques both in terms of prediction accuracy and interpretability. More broadly, our framework provides important insights into the physiological underpinnings of brain function.

## 1 Introduction

Connectomic studies use functional brain images of human subjects performing tasks to elucidate complex cognitive processes. Functional Magnetic Resonance Imaging (fMRI) is a common imaging modality used to analyze the underlying natural processes in healthy individuals and the dysregulation of such processes due to disease and/or injury. Functional networks derived from fMRIs typically superpose many neurophysiological responses elicited by stimuli. Identifying and separating functional networks into their basic building blocks is essential to explain the shared, and unique aspects of neuronal responses across heterogeneous populations performing different tasks. Ideally, these elemental networks should be grounded in neurophysiology, identifying coherent modules of neural responses that are interpretable by neuroscientists and other domain experts.

The method of choice for connectomic analysis is Independent Component Analysis (ICA) [36, 24], which is used on individual fMRIs to spatially localize regions of interest. Group-ICA [13, 10, 34] combines fMRIs across individuals to model shared regions of interest. Other ML-based interpretable methods have been proposed in the recent past [19, 26, 33, 29]. However, these methods are limited in their ability to handle large datasets with diverse subjects (young v/s old, healthy v/s diseased) performing a variety of cognitive tasks. Large-scale efforts, such as the Human Connectome Project [46], Cambridge Centre for Ageing and Neuroscience (Cam-CAN) dataset [42], and Alzheimer's Disease

Neuroimaging Initiative (ADNI) [25] have each collected and curated neuroimages from cohorts of several hundred subjects. Current efforts by the UK Biobank will image over 100,000 individuals [35]. Each of these datasets includes subjects of different ages, stages of neuroplasticity, and degeneration.

In this paper, we propose a novel framework that deconvolves networks derived from fMRIs of subjects performing different tasks into a small set of elementary networks that serve as building blocks that are: (i) shared across large cohorts; (ii) can be composed into task-specific networks; and (iii) are predictive of tasks and efficacy. We call these networks *canonical task connectomes*. Our framework also computes the extent of expression of these networks for each task, along with its neurophysiological basis.

Our approach first combines individual functional networks into a population-level matrix $\mathbf{X}$. We then deconvolve this matrix into its factors $\mathbf{A}$ and $\mathbf{S}$ such that each column $\mathbf{A}_{(i,*)}$ corresponds to a canonical task connectome, and the corresponding row $\mathbf{S}_{(*,i)}$ characterizes the extent to which the canonical network is expressed in every subject. However, since individual samples (fMRIs) correspond to subjects performing different tasks, the latent canonical representations must encode this important information. We accomplish this by formulating a suitable supervised matrix factorization problem, where factors are guided by a supervision matrix of tasks and subjects.

We present experimental results on the "unrelated set" of subjects in the Human Connectome Project. We compare results from two methods – Supervised Singular Value Decomposition (SupSVD) and Supervised Non-Negative Matrix Factorization (SupNMF) on subjects from HCP at rest and for six tasks (Language, Emotional Processing, Gambling, Motor, Relational Processing, and Social Processing). Our results show that:

- *Canonical task connectomes have high task-specificity*. We show that our approach constructs networks that uniquely characterize different tasks and are therefore excellent markers of tasks.

- *Canonical task connectomes are generalizable across cohorts*. We show that canonical representations obtained on a suitably constructed train set can accurately predict tasks being performed by the test set. We also show that SupNMF outperforms SupSVD in terms of prediction accuracy across ranges of parameters.

- *Canonical task connectomes identify common neural processes*. We show that our approach finds functional networks that are shared across tasks. This enables novel interpretations of processes and responses associated with different task stimuli.

- *Canonical task connectomes have a strong physiological basis*. We show that the canonical connectomes can be mapped to regions of the brain to identify physiological underpinnings of tasks, that are in strong agreement with literature in neurosciences.

The rest of the paper is organized as follows: in Sections 2.2 and 2.3, we discuss relevant methods for supervised matrix factorization. In Section 2.1, we provide details for our proposed framework. Then, we describe the HCP dataset and the preprocessing pipeline. This is followed by comprehensive experimental results in section 3, where we demonstrate the interpretability and generalizability of our proposed approach. Finally, we conclude with related methods in Section 4 and discussion in Section 5.

## 2 Methods and Materials

We describe our formulation and solution to the problem of identifying interpretable task-specific brain networks, called "connectomes" from neuroimaging datasets of subjects performing a variety of cognitive tasks. Connectomes are networks in which regions of the brain correspond to nodes and correlated activity quantifies the strength of edges across corresponding nodes (regions). We describe, in more detail, the process of construction of connectomes in Section 2.4.

We hypothesize and validate that neuronal activity observed during a task is composed of a small set of elementary patterns or motifs. Correspondingly, the observed connectome is a superposition of these motifs that we call canonical task connectomes. The goal of our formulation and methods is to demonstrate the existence and applications of such canonical task connectomes.

We abstract our connectome as a $region \times region$ similarity matrix. Our problem of finding canonical task connectomes can be formulated as one of Supervised Matrix Factorization (SMF) – a family of deconvolution techniques that expresses a data matrix as a sum of low-rank factors. The specific factors are determined by the optimization criteria and constraints associated with different methods. In

SMF, the factors are further guided by additional information (in our case, task labels associated with subjects) written as a *supervision matrix*. We use two state-of-the-art supervised matrix factorization techniques – Supervised Non-negative Matrix Factorization (SupNMF) and Supervised Singular Value Decomposition (SupSVD) to compute matrix factors. We suitably formulate our deconvolution problem for using these techniques, apply them to a large cohort of subjects, comprehensively compare their performance, and show that our formulation, combined with SupNMF yields highly interpretable, consistent, and strong task-specific signals.

## 2.1 Overview of our proposed framework

We write an observed connectome matrix $\mathbf{C}_O \in \mathbb{R}^{d \times d}$ as a linear combination of a small number of latent (i.e., unobserved) matrices $\mathbf{C}_l$.

$$\mathbf{C}_O^{(j)} = \sum_{i=1}^{r} \mathbf{S}^{(j,i)} \mathbf{C}_l^{(i)} \tag{1}$$

Here, $r$ denotes the number of latent connectomes (i.e., the dimensionality of latent space), $i$ denotes the index of latent connectome, $j$ is the index of observed connectome (subject or data sample) in the dataset, and $\mathbf{S} \in \mathbb{R}^{n \times r}$ represents the matrix of coefficients corresponding to the weights associated with each latent matrix. Each connectome (data sample) in the dataset has an associated task-label vector $y \in \{0,1\}^t$, where $t$ is the number of tasks. A connectome has exactly one non-zero in its label vector, corresponding to the task that was being performed by the subject during imaging. We aim to learn latent factors and use them to construct a predictor $f$ that takes in a row of $\mathbf{S}$ and correctly predicts the task.

$$\hat{y}^{(j)} = f(\mathbf{S}^{(j,*)}) \tag{2}$$

Here, $f : \mathbb{R}^r \to \{0,1\}^t$. Combining equations 1 and 2, we get our objective function

$$\underset{\mathbf{S},f,\mathbf{C}_l}{\text{minimize}} \quad \sum_{j=1}^{n} \left( ||\mathbf{C}_O^{(j)} - \sum_{i=1}^{r} S^{(i,j)} \mathbf{C}_l^{(i)}||_F^2 + \lambda(y^{(j)} - f(\mathbf{S}^{(j,*)})))^2 \right) \tag{3}$$

Here, $n$ denotes the total number of data samples, and $d$ denotes the number of regions in each connectome. The first term in the objective function minimizes the approximation error, and the second term minimizes the classification error. The relative importance of the two terms are controlled by the tuneable parameter $\lambda$. Rather than working with tensors, we simplify our setting by: a) vectorizing the connectomes and stacking them column-wise into a population-level data matrix, $\mathbf{X} \in \mathbb{R}^{\mathcal{O}(d^2) \times n}$; and b) modeling $f(.)$ as a linear function. We create a one-hot matrix $\mathbf{Y}$ to represent labels for the different cognitive tasks performed by the subjects. We now model the problem as one of supervised matrix factorization. We compute the factors using different matrix factorization techniques – NMF, SVD, SupNMF, and SupSVD. We discuss the latter two approaches in Sec 2.2 and 2.3 respectively. On this matrix, we note that the columns of the basis matrix are connectomes that are superposed to approximate of the columns of $\mathbf{X}$. We call them "canonical task connectomes". Our results show that these representations strongly correlate with anatomical and physiological processes associated with different tasks.

To show the generalizability of these canonical representations, we divide the cohort randomly into train and test sets. We use the canonical representation computed from the train set to infer cognitive tasks performed by subjects in the test set. In Fig. 1, we illustrate the general framework. Using the train set, we find a small number of canonical task connectomes that serve as basis vectors to explain brain activity in a large cohort. For the test set, we find coefficients that best fit the previously computed basis. Next, we learn a model to map coefficients in the train set to labels. We use this model on the test coefficients to predict tasks performed.

## 2.2 Supervised Non-negative Matrix Factorization

Let $\mathbf{X} \in \mathbb{R}_{\geq 0}^{p \times n}$ denote the data matrix, $\mathbf{Y} \in \mathbb{R}_{\geq 0}^{k \times n}$ denote a class label matrix, and $k$ the number of different classes. Supervised Non-negative Matrix Factorization (SupNMF) is defined as:

$$\underset{\mathbf{A},\mathbf{S},\mathbf{B} \geq 0}{\arg\min} ||\mathbf{X} - \mathbf{AS}||_F^2 + \lambda||\mathbf{Y} - \mathbf{BS}||_F^2 \tag{4}$$

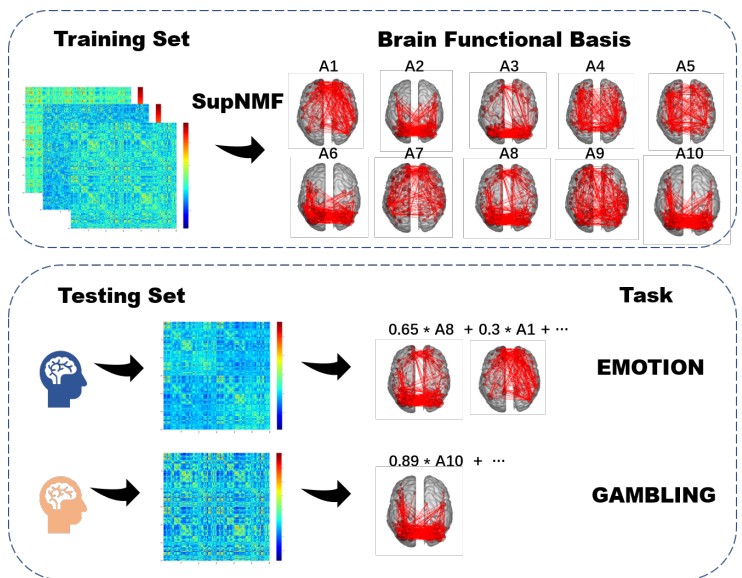

Figure 1: *Overview of our framework:* (1) The training phase deconvolves the data matrix of vectorized connectomes in the training set into a small number of basis vectors; (2) The testing phase computes the coefficients of the functional basis and predicts the task on new subjects.

Here, $\mathbf{A} \in \mathbb{R}_{\geq 0}^{p \times r}$ is the (non-negative) "basis matrix" which is a low-rank, latent description of the columns of $\mathbf{X}$, $\mathbf{S} \in \mathbb{R}_{\geq 0}^{r \times n}$ is the (non-negative) matrix of coefficients that provides the weights to each of the latent factors required to explain each data-point, $\mathbf{B} \in \mathbb{R}_{\geq 0}^{k \times r}$ is the matrix that minimizes classification error, and $\lambda$ controls the relative importance of the supervision term. The first term minimizes the error in reconstructing the data matrix and the second term minimizes classification error.

A few points to note: a) When $\lambda = 0$, this formulation reduces to unsupervised Non-negative Matrix Factorization [47] [31]; b) objectives such as information divergence can be used in lieu of the Frobenius Norm [22]. This problem has been modeled as a block multi-convex problem by Haddock et al. [22] to derive the following algorithm with multiplicative updates. In the algorithm, $\mathbf{M} \odot \mathbf{N}$ represents the Hadamard Product (i.e., element-wise product) of matrices $\mathbf{M}$ and $\mathbf{N}$. Similarly $\frac{\mathbf{M}}{\mathbf{N}}$ represents Hadamard Division.

---

**Algorithm 1:** Supervised NMF

---

**Input:** $\mathbf{X}, \mathbf{Y}, r, \lambda, N$
**Initialize:** $\mathbf{A} \in \mathbb{R}_{\geq 0}^{p \times r}, \mathbf{B} \in \mathbb{R}_{\geq 0}^{k \times r}, S \in \mathbb{R}_{\geq 0}^{r \times n}$
**for** $i = 1, ..., N$ **do**

$\quad \mathbf{A} \leftarrow \mathbf{A} \odot \dfrac{\mathbf{X}\mathbf{S}^T}{\mathbf{S}\mathbf{S}^T}$

$\quad \mathbf{B} \leftarrow \mathbf{B} \odot \dfrac{\mathbf{Y}\mathbf{S}^T}{\mathbf{B}\mathbf{S}\mathbf{S}^T}$

$\quad \mathbf{S} \leftarrow \mathbf{S} \odot \dfrac{\mathbf{A}^T\mathbf{X} + \lambda\mathbf{B}^T\mathbf{Y}}{\mathbf{A}^T\mathbf{A}\mathbf{S} + \lambda\mathbf{B}^T\mathbf{B}\mathbf{S}}$

**end**

---

## 2.3 Supervised SVD

Supervised Singular Value Decomposition (SupSVD) [32] incorporates a supervision matrix into conventional SVD. It assumes that the data matrix $\mathbf{X} \in \mathbb{R}^{n \times p}$ contains latent, low-rank information

that is shared with the supervision matrix $\mathbf{Y} \in \mathbb{R}^{n \times k}$. The SupSVD model can be expressed as follows:

$$\mathbf{X} = \mathbf{U}\mathbf{V}^T + \mathbf{E}$$
$$\mathbf{U} = \mathbf{Y}\mathbf{B} + \mathbf{F} \tag{5}$$

Here, $\mathbf{U} \in \mathbb{R}^{n \times r}$ is a latent score matrix, $\mathbf{V} \in \mathbb{R}^{p \times r}$ is a full-rank loading matrix, and $\mathbf{B} \in \mathbb{R}^{k \times r}$ is a coefficient matrix, with $\mathbf{F} \in \mathbb{R}^{n \times r}$ and $\mathbf{E} \in \mathbb{R}^{n \times p}$ being error matrices. For model estimation, a modified version of the expectation–maximization (EM) algorithm was proposed by Li et al. [32].

## 2.4 Data

We validate our model and methods on data from the Human Connectome Project (HCP) Young Adult dataset [46]. Specifically, we use the fMRIs from the set of 100 "unrelated subjects". For each subject, we have separate fMRIs when they are at rest, and while they perform six cognitive tasks (Language, Emotional Processing, Gambling, Motor, Relational Processing, and Social Processing) [2]. We first use the Minimal Pre-Processing Pipeline prescribed by the HCP consortium [18]. This process includes spatial artifact/distortion removal, head motion correction, registration, and normalization to standard space. For each input noisy fMRI, the Minimal Preprocessing Pipeline outputs a clean and standardized $voxel \times time$ time-series matrix. Then, we use the Atlas of Glasser et al. [17] to aggregate this matrix into a $region \times time$ matrix. We note that each *region* consists of proximal *voxel*s with shared anatomy. In all, the Glasser Atlas demarcates 180 regions in each hemisphere (360 in total). We then create the functional connectome (FC) matrix for each fMRI by computing the Pearson Correlation between all pairs of regions. In all, we have 700 FCs (100 $subjects \times 7$ $tasks$). We vectorize the upper triangular matrix of each FC and stack them side by side to create a population-level matrix of size $700 \times 64620$.

# 3 Results

In this section, we show that our canonical task connectomes are highly specific to a small subset of tasks, and as a consequence provide both an understanding of the neural response, as well as the ability to predict tasks. Then, we provide evidence of strong spatial localization for these representative brain networks, which establishes interpretability on the basis of neuro-anatomy. We also show that the regions implicated in the tasks are supported by prior experimental studies, which establishes physiological interpretations.

## 3.1 Canonical Task Connectomes have High Task Specificity

In the first set of results, we demonstrate that our connectomes encode information that is unique to each task. This is non-trivial because of: a) inherent heterogeneity in basal brain activity across individuals; b) individual-level differences in cognitive processes to perform a task; c) diversity of task conditions; and d) noise in the imaging modality. Using four methods for matrix factorization – SupNMF, NMF, SupSVD, and SVD, we deconvolve the population-level matrix $\mathbf{X}$ to find different sets of canonical task connectomes and the corresponding linear coefficients that quantify the extent to which each canonical task connectome is present in every functional connectome. For the purposes of visualization, we project the coefficients' matrix ($\mathbf{S}$ for NMF/SupNMF, and $\mathbf{U}$ for SVD/SupSVD) into two dimensions using UMAP, shown in Fig 2. We observe that in each case, resting-state (Red) FCs are always clustered separately. This confirms that resting-state brain activity is very different from all task-specific brain activity. We also see that Language (Blue) and Social Processing (Purple) are also clearly separated by all four methods. This suggests that the task-specific networks in the case of these two tasks are strongly distinct, and can be deconvolved with no supervision.

However, other tasks – Emotion Processing (Green), Gambling (Amber), Motor (Pink), and Social Processing (Gray) are separated only by SupNMF (Fig 2a. The lack of separation observed in NMF, SVD, and SupSVD strongly indicates that the canonical representations obtained by SupNMF are most task-specific. To quantify the task discriminatory power of our approach, we cluster the coefficients using k-means for $k = \{1,...,7\}$ and compute the Adjusted Rand Index (ARI) in each case. The results are shown in Fig 3. It is evident that ARI for SupNMF is significantly higher for all choices of $k$. For NMF, SVD, and SupSVD, ARI plateaus at $k = 4$, which is consistent with the UMAP plots.

Since Canonical Task Connectomes are shown to be task-specific, they provide excellent representations to classify tasks performed by a test subject.

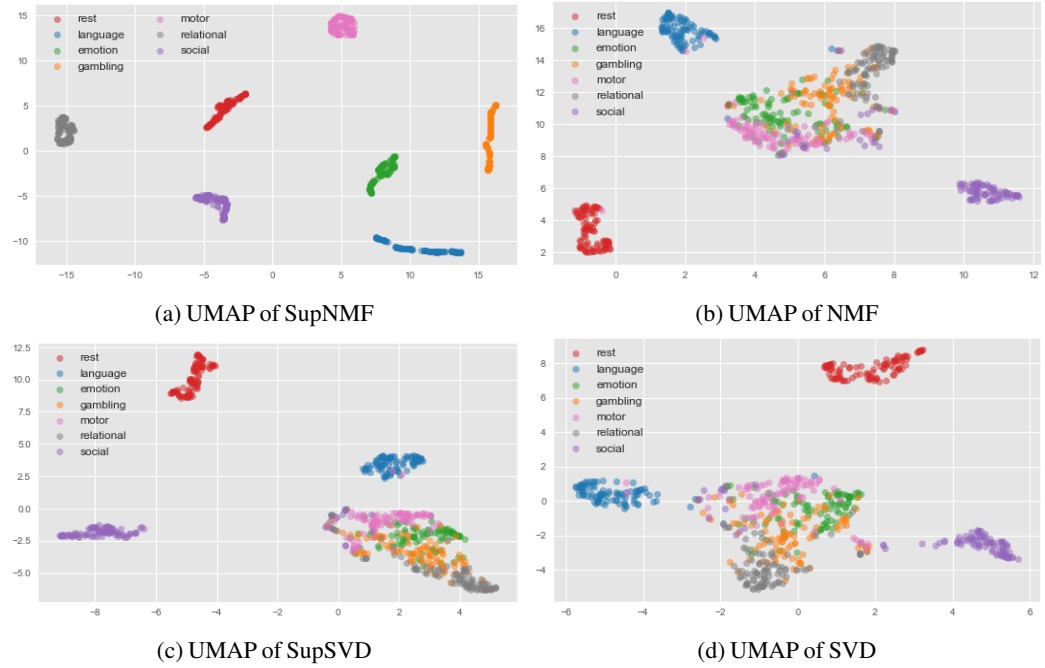

(a) UMAP of SupNMF

(b) UMAP of NMF

(c) UMAP of SupSVD

(d) UMAP of SVD

Figure 2: *Task-specificity of canonical task connectomes obtained by different methods.* We use UMAP to visualize the "coefficients matrix" for different tasks. (a)-(d) show the results for SupNMF, NMF, SupSVD, and SVD respectively. We see that Rest (Red), Language (Blue), and Social (Purple) are clustered in all four cases. However, the remaining tasks are only separated by SupNMF.

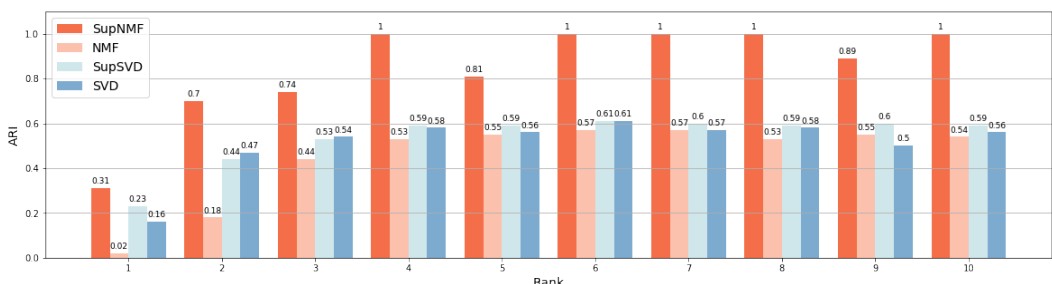

Figure 3: *ARI values for k-means clustering on the coefficients obtained by SupNMF, NMF, SupSVD, and SVD.* We observe that ARI for SupNMF is consistently higher than other methods.

### 3.2 Canonical Task Connectomes are Generalizable across Cohorts

We show that canonical task connectomes are stable representations of different tasks. To demonstrate this, we first compute the canonical representations on a training set. Then, we predict the task performed in the test-set. More specifically, we create $\mathbf{X}_{train}$ and $\mathbf{X}_{test}$ by 80/20 random splits of the subjects. We deconvolve $\mathbf{X}_{train}$ to find the canonical task connectomes $\tilde{\mathbf{A}}$ and the coefficient matrix $\tilde{\mathbf{S}}$, and use $\tilde{\mathbf{S}}$ along with corresponding task labels to train a classifier. Now, given a test subject (or test set), we compute the least-squares solution $\hat{\mathbf{S}}$ using $\hat{\mathbf{S}} = \tilde{\mathbf{A}}^{\dagger} \mathbf{X}_{test}$. Finally, we predict the labels of $\mathbf{X}_{test}$ using $\hat{\mathbf{S}}$ and the trained classifier.

We compare the test accuracy of SVD, NMF, SupNMF, and SupSVD using three classifiers – K-nearest neighbor, support vector machine, and a 3-layer perceptron. In Table 1, we summarize the results for rank-10 approximations, averaged across 10 runs. The factors computed by SupNMF yield high accuracy (>88%) for all three classifiers. The factors output by SupSVD and NMF also perform well in predicting task conditions. This can be attributed to the fact that while individual factors of SupSVD and NMF are not task-specific, the combinations of different factors still have reasonable predictive

Table 1: *Test accuracy using different classifiers*

| Method | SupNMF | SupSVD | NMF | SVD |
|--------|--------|--------|-----|-----|
| KNN | **88.54 ± 0.49** | 83.30 ± 2.00 | 82.64 ± 2.02 | 69.11 ± 4.24 |
| MLP | **88.14 ± 2.16** | 83.96 ± 2.53 | 87.36 ± 2.33 | 74.44 ± 3.98 |
| SVM | **87.64 ± 2.00** | 86.09 ± 3.31 | 86.86 ± 1.70 | 73.61 ± 2.73 |

power. This is evident from Fig. 4, where we show the normalized and thresholded columns of $\tilde{S}$ from both SupNMF and NMF. In SupNMF, most connectomes from a common task are assigned to the same "canonical connectome". However, in NMF, we see that connectomes from a common task are assigned to different factors. The accuracy of predicting on the basis of singular vectors is poor, due to the orthogonality constraints enforced on the columns of $U$. We show similar plots for SupSVD and SVD in Supplementary Section 1.

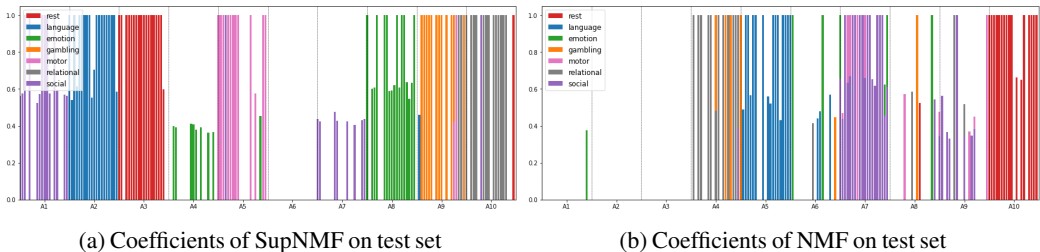

(a) Coefficients of SupNMF on test set       (b) Coefficients of NMF on test set

Figure 4: *Coefficient matrices of SupNMF and NMF for test connectomes.* We L-1 normalize the columns of $\tilde{S}$ obtained by both SupNMF and NMF and fit to the Normal Distribution. We then use 90 percentile as the cutoff to discard small values in both matrices. For coefficients in SupNMF, we see that each "canonical connectome" is assigned to one task in most cases. This is evident by the minimal mixing of colors. However, in NMF, we see that coefficients corresponding to a common task are spread across different "canonical connectomes". In this figure, each task(color) has 20 connectomes. We compute rank-10 coefficient matrices in both cases.

### 3.3 Canonical Task Connectomes have a Strong Anatomical and Physiological Basis

We show that: a) each canonical task connectome is spatially localized to anatomically demarcated lobes; and b) the regions enriched in each canonical connectome are known to be implicated in the corresponding task. As before, we deconvolve the population-level matrix $X$ to compute $A$ and $S$. In this experiment, we use rank 20 approximation to aid in the interpretation. From each column $A_{(*,i)}$, we construct $region \times region$ canonical task connectome $C_i$. Finally, we create adjacency matrices by retaining the top 5% of edges from $C_i$.

In Fig 6, we visualize the task-specific connectomes. We restrict our analysis to nodes with $degree > 35$ (p-value < 1e-5). We note that edges containing the Prefrontal Cortex are over-represented in A4, A5, A7, and A18; MotorStrip is over-represented in A17; Parietal Lobe is over-represented in A2, A4, and A20; and the Occipital Lobe is over-represented in A1, A2, and A9. In each case, the observation is statistically significant with p-values < 1e-10. We note that the temporal lobe is the only major region not represented in these canonical connectomes. In all, this high degree of spatial locality indicates a strong anatomical basis.

Next, we normalize the columns of $S$ given by SupNMF and fit to a Gaussian and retain only those non-zero values higher than 90 percentile. In Fig. 5, the rows of $S$ are visualized in a combined graph. It is evident that the non-zeros of these significant coefficients are highly selective of tasks. In fact, coefficients are active only for one specific task. With the knowledge of both the anatomical basis of each canonical connectome (Fig. 6 and their associated tasks (Fig. 5), we can now establish the physiological basis for these canonical connectomes. We find that our method finds patterns that are supported by neuroscience experiments reported in literature. Regions in the left prefrontal cortex are associated with word and sentence comprehension [16], which is over-represented in A4 of Fig 6 corresponding to the language task, as shown in S4 of 5. The dorsal Default Mode Network (dDMN) is known to be active during Rest [5]. The anatomical regions for this functional network to the posterior

cingulate cortex (in the limbic node), and the angular gyrus found in the posterior part of the inferior parietal lobe, which is over-represented in A9 of Fig 6. Additionally substructures corresponding to the dorsal medial prefrontal cortex are also found in A9. We see that rest connectomes are strongly activated for the corresponding column in the **S** matrix, as shown in Figure 4. The regions implicated in social processing are the medial prefrontal cortex, which is located in the prefrontal cortex of the frontal lobe [15]. In our results, these nodes are over-represented in A18. Finally, the regions implicated in relational processing are dorsolateral Prefrontal Cortex rostrolateral prefrontal cortex and posterior parietal cortex [23]. These regions are over-represented in A20, and A3 respectively.

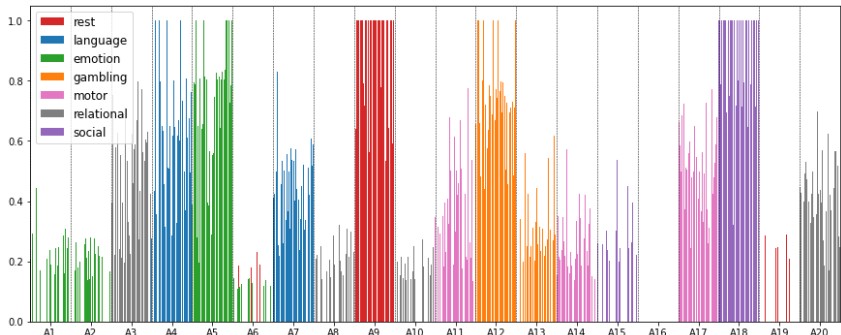

Figure 5: *Coefficients matrix* **S** *of SupNMF.* We normalize the columns of **S** and fit to the Normal Distribution. We then use 90 percentile as the cutoff to discard small values. Each row of **S** is nearly exclusive to one task (as indicated by the minimal mixing of colors/ tasks). We combined all rows of **S** into one plot for effective visualization (by summing across columns). The colors code for rest and six tasks. Within each rectangle bounding box, we have the entire cohort of 100 subjects .

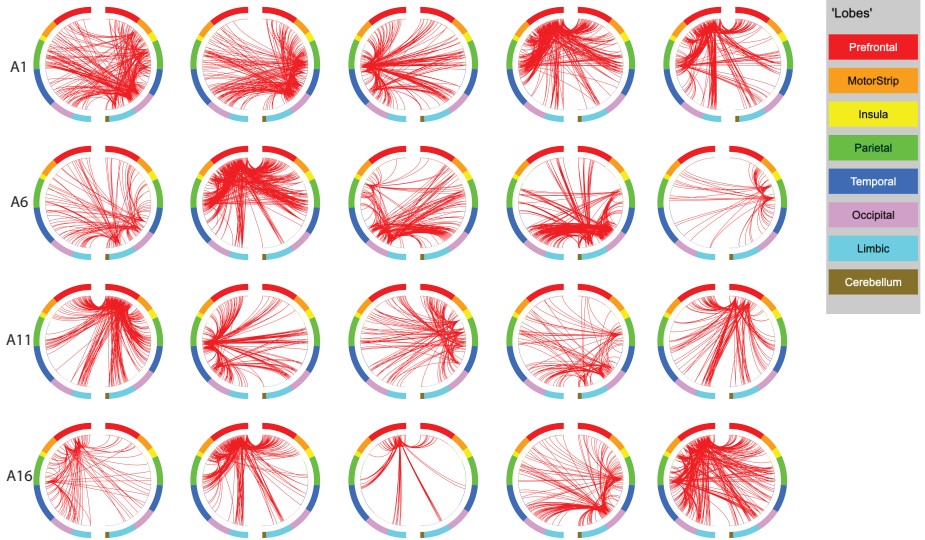

Figure 6: *Canonical Task Connectomes have strong anatomical basis.* In each of the 20 canonical task connectomes, the disconnected semi-circles represent the two hemispheres. Each dot in the inner side of these hemispheres corresponds to a micro-region in the brain. In all, there are 360 micro-regions, each can be mapped directly to one of the coarser lobes. The connectomes shown here have strong spatial localization. As an example, nearly all edges in A1 and A2 have one end in the Right Occipital Lobe. This visualization uses the BioimageSuite [37]

## 4 Related Work

**Matrix Factorization** Independent Component Analysis (ICA) and its variants are widely used in fMRI analysis. Spatial Independent Component Analysis (ICA) [36, 48, 40, 7, 6] methods decompose fMRI data into a set of spatially independent components. They identify patterns of activity across the brain that are independent of one another. This information is used to identify distinct networks of brain regions involved in various cognitive processes. In a typical ICA model, the source signals are assumed to be statistically independent and non-Gaussian, with an unknown linear mixing process. The model assumes that every observed vector $x \in \mathbb{R}^m$ is generated by a linear mixture of $n$ independent sources $x = \mathbf{A}s$, where $s \in \mathbb{R}^n$ is an N-dimensional vector whose elements are the random variables that refer to the independent sources and $\mathbf{A} \in \mathbb{R}^{m \times n}$ is an unknown mixing matrix. ICA aims to estimate an unmixing matrix $\mathbf{W} \in \mathbb{R}^{n \times m}$ such that the recovered sources: $y = \mathbf{W}x = \mathbf{W}\mathbf{A}s$ is a good representation of the true sources $s$. Applying the typical ICA model to fMRI data, we have data $\mathbf{X} = \mathbf{A}\mathbf{S}$, where $\mathbf{X} \in \mathbb{R}^{N \times V}$ spans $N$ time points and $V$ voxels, and $\mathbf{S}$ contains spatially independent source signals. Group ICA is an extension of spatial ICA that allows the identification of common patterns of activity across multiple subjects in a study. A popular implementation of Group ICA is Multivariate Exploratory Linear Decomposition into Independent Components (MELODIC) [3], which is part of the fMRI Standard Library (FSL). Other approaches for multi-subject analysis using ICA have been proposed [8, 14, 20, 38, 4]. The model in Calhoun et al. [8] defines Group ICA as $\mathbf{X}_i = \mathbf{A}_i\mathbf{S}$, where $\mathbf{X}_i \in \mathbb{R}^{N_i \times V}$ is the fMRI observation for subject $i$ with $N_i$ time points and $V$ voxels. Group ICA captures a group subspace with independent spatial maps and time courses. Then, these are used to reconstruct subject-specific spatial maps $\mathbf{S}_i$ and time courses $\mathbf{A}_i$. Group ICA has been widely used to study functional connectivity differences between groups of healthy and clinical populations [43, 11], as well as to identify brain networks associated with specific cognitive processes across a group of individuals [12, 28]. However, both Spatial and Group ICA are limited as they are unsupervised approaches that find dominant patterns in the entire dataset. This comes at the expense of ignoring more intricate patterns such as: a) differences across individual subjects; and b) shared patterns with subsets of subjects (such as disease, cognitive tasks, etc). Since the "canonical task connectomes" computed in our approach are guided by additional information relating to subjects/ samples (such as task or disease labels), our approach is more flexible and powerful than traditional approaches.

**Other interpretable methods** Subspace clustering methods are used in fMRI to partition data into subspaces and assign each data point (e.g., voxel or region of interest) to its corresponding subspace. This allows for the identification of different brain activity patterns or functional connectivity profiles within data. Several subspace clustering methods have been applied to fMRI data such as spectral clustering [21, 9, 1], sparse subspace clustering [41, 30], low-rank and sparse decomposition (LRSD) [44, 45, 39]. Subspace clustering reveals distinct brain activity patterns, functional networks, or connectivity profiles within the data. However, there are some key limitations of subspace clustering including its unsupervised nature, reliance on unlabeled data, limited generalization to new datasets, and challenges in interpreting identified subspaces. In contrast, we demonstrate that our method generates task-specific feature representations, is generalizable, and facilitates interpretation by domain experts. Graph Neural Networks and other Deep Neural Network models have also been used to identify regions of interest (and functional networks) shared across a cohort of subjects [33, 29, 27]. However, these methods cannot separate the distinct networks, which limits their applicability to our problem. Our framework explains observed (composite) brain activity in terms of elementary networks, which have biological basis.

## 5 Conclusion

We presented a new problem and framework for fMRI analysis that deconvolves an input set of neuroimages of subjects performing different cognitive tasks into a compact set of task-specific elementary networks called "canonical task connectomes". We formulate our problem as one of supervised matrix factorization and show that the resulting latent factors/ networks can be interpreted as "building blocks" for the different cognitive tasks. We show experimental results on the Human Connectome Project dataset, which demonstrate that SupNMF captures the natural task-specific structure in suitably abstracted neuroimages. We also show that these canonical task connectomes are useful biomarkers to predict the task being performed. Additionally, we show anatomical and physiological underpinnings for the networks identified by our framework.

Our framework can be extended to more complex applications, such as: a) understanding shared and unique functional networks across different pathologies; and b) how task-specific networks can get dysregulated due to the onset, and progression of diseases.

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
