# Supplementary Material: Deconvolving Complex Neuronal Networks into Interpretable Task-Specific Connectomes

## A    Canonical Task Connectomes are Generalizable across Cohorts

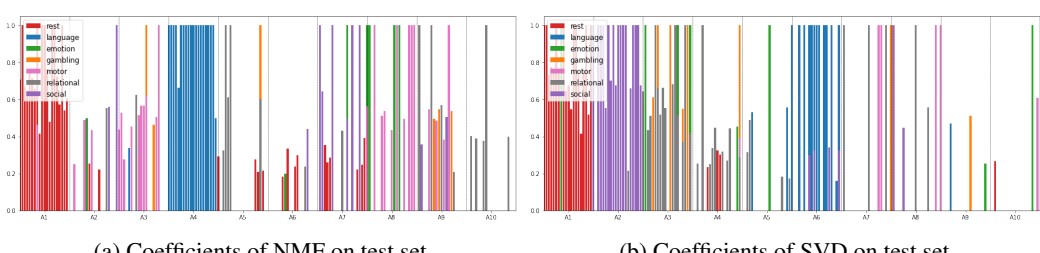

(a) Coefficients of NMF on test set        (b) Coefficients of SVD on test set

Figure 1: *Coefficient matrices of SupNMF and SupSVD for test connectomes.* We L-1 normalize the columns of Ũ obtained by SupSVD and SVD and fit to the Normal Distribution. We then use 90 percentile as the cutoff to discard small values in both matrices. For coefficients in SupSVD, we see that tasks associated with each "canonical connectome" are better separated than with SVD. In this figure, each task(color) has 20 connectomes. We compute rank-10 coefficient matrices in both cases.

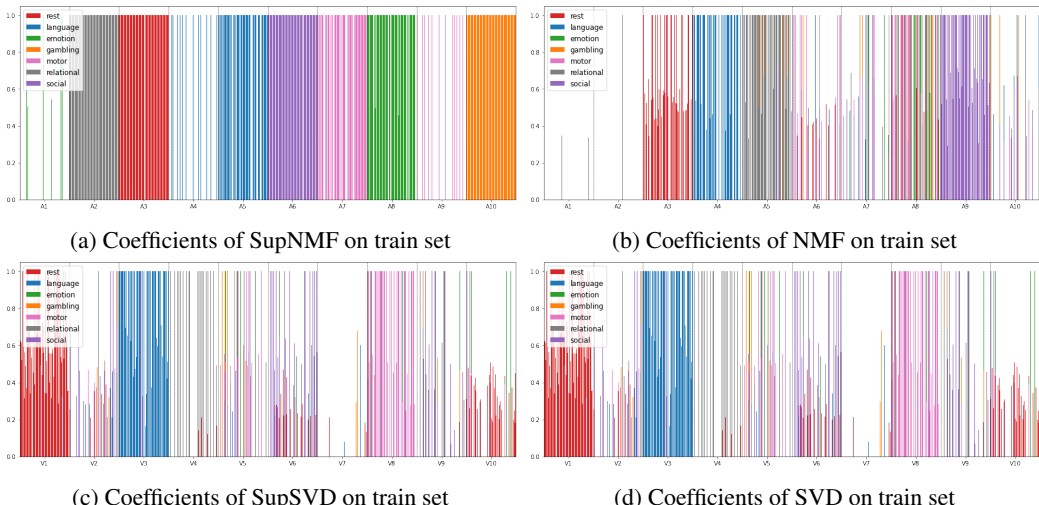

(a) Coefficients of SupNMF on train set

(b) Coefficients of NMF on train set

(c) Coefficients of SupSVD on train set

(d) Coefficients of SVD on train set

Figure 2: *Coefficient matrices of SupNMF, NMF, SupSVD and SVD for train connectomes.* We L-1 normalize the columns of $\tilde{\mathbf{S}}$ obtained by SupNMF and NMF, and of $\tilde{\mathbf{U}}$ obtained from SupSVD and SVD. Then, we fit to the Normal Distribution. We use 90 percentile as the cutoff to discard small values in both matrices. We observe that even on the train set, SupNMF results in the best separation of tasks, which is evident from the minimal mixing of colors. In this figure, each task(color) has 20 connectomes. We compute rank-10 coefficient matrices in all cases.