# OpenReview forum: "Deconvolving Complex Neuronal Networks into Interpretable Task-Specific Connectomes"
_NeurIPS.cc/2023/Conference — Submitted to NeurIPS 2023_

### Official Review · Reviewer_1xzo · 2023-07-05

**Soundness:** 2 fair
**Presentation:** 2 fair
**Contribution:** 2 fair
**Rating:** 4
**Confidence:** 4

**Summary:**

This paper addresses the challenge of identifying elementary functional neuronal networks and their combinations in the context of complex tasks, using task-specific functional MRI (fMRI) data. The central problem it tackles is the deconvolution of task-specific aggregate neuronal networks into elementary networks. These elementary networks can then be used for functional characterization and mapped to underlying physiological regions of the brain. Due to the high-dimensionality, small sample size, acquisition variability, and noise inherent in this task, the authors propose a deconvolution method based on supervised non-negative matrix factorization (SupNMF). The results demonstrate that SupNMF can uncover cognitive "building blocks" of task connectomes that are physiologically interpretable, predict tasks with high accuracy, and outperform other supervised factoring techniques in both prediction accuracy and interpretability. Overall, the proposed framework offers valuable insights into the physiological foundations of brain function and individual performance.

**Strengths:**

1. The paper presents a valuable effort to implement a supervised decomposition method in a novel way, showing the potential for this approach in a complex context, such as the analysis of neuronal networks.
2. The authors provide fascinating results indicating that each task has unique markers within these learnable networks. This insight could contribute significantly to understanding how tasks are represented and processed within the brain. Observing that some networks are shared across tasks also provides a meaningful direction for future research.
3. The alignment of the findings with existing physiological research is a good sense for future study.

**Weaknesses:**

1. Reproducibility: The study could be enhanced by applying the proposed method to other datasets or by resampling the existing dataset. This would help to assess the generalizability of the method and the robustness of the findings, which is currently a limitation of the work.
2. Baseline Comparison: It would be beneficial if the authors had compared the proposed SupNMF method with other supervised decomposition methods, such as Partial Least Squares regression. This lack of comparison limits the understanding of how their proposed method stands in relation to existing methodologies regarding performance and effectiveness.
3. Parameter Study: The authors need to provide an in-depth analysis or sensitivity study concerning the weight parameter \lambda. As this parameter likely plays a significant role in balancing different loss terms, this omission constitutes a substantial weakness, potentially leaving readers unclear about the effectiveness of supervision signals.

**Questions:**

1. In the abstract, line 9 uses the wrong left double quotation mark and right double quotation mark.
2. Provide a reference and explanation about UMAP mentioned in the paper. For the benefit of readers who might not be familiar with this method, I recommend that the authors reference a key source on UMAP and provide a brief explanation of its use and significance in this context.
3. In the related work, there is a missing related paper when talking about the interpretable GNN, "Interpretable Graph Neural Networks for Connectome-Based Brain Disorder Analysis. MICCAI 2022"

**Limitations:**

Yes

---

> ### Author Rebuttal · Authors · 2023-08-09
>
> We thank the reviewer for the detailed review and helpful comments.
>
> Weakness:
>
> 1.Reproducibility: The study could be enhanced by applying the proposed method to other datasets or by resampling the existing dataset. This would help to assess the generalizability of the method and the robustness of the findings, which is currently a limitation of the work.
>
> **A**: We include additional results on a new dataset from the Cambridge Center for Aging and Neuroscience (CamCAN) segmented using a Harvard Oxford Atlas, as suggested by the reviewer. Our results show that the proposed method is consistently highly accurate on this new dataset as well – please see included results in the rebuttal PDF.
>
> Furthermore, we add additional results on the robustness of our method by examining our results across the full range of number of factors from 7 to 20 (7 because we have 7 tasks in HCP and 20 because the dominant factors saturate by rank 20). Our results are remarkably robust across this range of number of factors. We thank the reviewer for the suggestion – the suggested experiments further highlight the power of our method.
> Table 1: Test accuracy using different sample size and rank on HCP dataset
>
> |rank = 20| SupNMF|||NMF|||
> |-|-|-|-|-|-|-|
> |subjects |20|50|100|20|50|100|
> |KNN|83.79±2.41|90.97±1.26|88.7±0.84|76.07±9.85|76.29±3.79|83.5±1.85|
> |MLP|86.96±5.68|89.36±3.11|88.39±2.85|81.07±5.77|86.43±2.58|88±2.34|
> |SVM|88±3.26|89.18±2.37|88.79±2.17|66.43±6.43|84.14±2.51|87±2.8|
>
> | rank = 15 | SupNMF|||NMF|||
> |-|-|-|-|-|-|-|
> |subjects|20|50|100|20|50|100|
> |KNN|81.5±2.78|88.24±0.99|88.79±0.67|56.43±9.29|82.43±4.65|82.43±3.5|
> |MLP|80.54±7.27|84.43±4.93|87.29±3.4|78.57±8.45|83.71±4.87|87.14±2.26|
> |SVM|88.21±7.67|88.29±3.31|89.5±1.24|73.93±7.33|76.29±6| 86±2.62|
>
> | rank = 10 | SupNMF|||NMF|||
> |-|-|-|-|-|-|-|
> |subjects |20|50|100|20|50|100|
> | KNN  | 83..71±1.91 | 85.72±1.44 | 88.54±0.49 | 70.71±4.74  | 75.14±4.05 | 82.64±2.02 |
> | MLP  | 81.43±7.46  | 88.5±3.92  | 88.14±2.16 | 72.5±8.3    | 83.43±4    | 87.36±2.33 |
> | SVM  | 84.64±7.67  | 88.5±2.83  | 87.64±2.16 | 71.79±11.46 | 76.43±2.49 | 86.86±1.7  |
>
> As shown in the table, SupNMF demonstrates consistent performance across varying subject numbers and ranks. In contrast, competing baselines such as NMF experience a significant drop in accuracy with fewer subjects.
>
> Accompanying visualizations illustrating task accuracy spanning ranks 7 through 20 and varying subject counts are provided in the author rebuttal PDF. This enhanced analysis will be incorporated into the final version of the paper.
>
> 2.Baseline Comparison: It would be beneficial if the authors had compared the proposed SupNMF method with other supervised decomposition methods, such as Partial Least Squares regression. This lack of comparison limits the understanding of how their proposed method stands in relation to existing methodologies regarding performance and effectiveness.
>
> **A**: Partial Least Squares (PLS) regression optimizes the covariance between the predictors and the response. PLS does not attempt to derive interpretable results, which is the primary motivation for our work. The high classification accuracy of our method is an added benefit. For this reason, we compare primarily against state of the art methods capable of delivering interpretable results (the best known current method in the class is SupSVD, which we have used as our primary baseline). In response to the reviewer, we have added another baseline, ICA, which is the method of choice in the neurosciences community. Our results show that the proposed method consistently outperforms ICA in classification accuracy, while yielding physiologically interpretable results (factors).
>
> 3.Parameter Study: The authors need to provide an in-depth analysis or sensitivity study concerning the weight parameter \lambda. As this parameter likely plays a significant role in balancing different loss terms, this omission constitutes a substantial weakness, potentially leaving readers unclear about the effectiveness of supervision signals.
>
> **A**: We set regularization parameters relative to the scale of the input data, specifically, lam=100\*np.linalg.norm(X,'fro'), reflecting the Frobenius norm of the matrix Xt. This ensures that the regularization is meaningful in the context of the data's magnitude. Parameter selection could also rely on heuristic techniques or inherent information. However, our default parameter selection works well in all of our experiments. Stated otherwise, our method does not need any parameter tuning, which is a significant benefit.
>
> We present results for a broad range of lambda values in the table below, demonstrating our parameter choice yields optimal outcomes.
>
> |lambda|1|10|10^2|10^3|10^4|100*np.linalg.norm(X_train.T,'fro')|
> |-|-|-|-|-|-|-|
> |test acc|76.36±4.16|80±3.19|81.79±2.73|87.5±2.29 |88.43±2.41|88.71±2.45|
>
> We now address the specific questions raised by the reviewer in the following.
>
> **Q1**: In the abstract, line 9 uses the wrong left double quotation mark and right double quotation mark.
>
> **A1**: We will fix this in the revised submission.
>
> **Q2**: Provide a reference and explanation about UMAP mentioned in the paper. For the benefit of readers who might not be familiar with this method, I recommend that the authors reference a key source on UMAP and provide a brief explanation of its use and significance in this context.
>
> **A2**: UMAP, akin to t-SNE is a very commonly used visualization tool in the community. Space constraints limited a larger explanation; however, the revised version will duly incorporate relevant references and explanation.
>
> **Q3**:In the related work, there is a missing related paper when talking about the interpretable GNN, "Interpretable Graph Neural Networks for Connectome-Based Brain Disorder Analysis. MICCAI 2022"
>
> **A3**: We appreciate the citation and will add it with the observation that related techniques have been used in the context of brain disorder analysis.

---

> > ### Comment · Reviewer_1xzo · 2023-08-14
> >
> > Thanks for the detailed response. I will increase my rating for this paper

---

### Official Review · Reviewer_zaHM · 2023-07-05

**Soundness:** 3 good
**Presentation:** 3 good
**Contribution:** 2 fair
**Rating:** 4
**Confidence:** 4

**Summary:**

This paper presents a decomposition method for task-functional connectivity. It proposes canonical task connectomes which derives sub-structure of functional brain connectivity which group connectomes which identify elementary components of the overall connection. The authors use supervised non-negative matrix factorization to factor connectome matrices and show that the derived features are suitable to predict tasks for functional MRI and robust dimension reduction of the original representation.

**Strengths:**

- The paper is clearly written.
- It construct a clear optimization problem whose results are easily interpretable.
- It demonstrates a solid dimension reduction for connectome data.

**Weaknesses:**

- Motivation for including supervision in the connectome decomposition is very weak.
- There are some missing details on variable descriptions, e.g., $d$ in line 96 is missing, $\hat y$ in eq (2) is missing (although can be inferred that it is a prediction for a class), and derivation of $X$ from $C$ should be better explained.
- Experiment is performed only on one study. It can be excused if the dataset is rare, but there are so many publicly available fMRI data.
- Lack of baselines. It is missing the most fundamental baseline, i.e., LDA. Moreover, just typing in "supervised dimension reduction" in google scholar yields various literature but this paper demonstrates only SupSVD as a supervised baseline.



**Questions:**

- I am not sure what the role of supervised SVD is in this paper. It is written as a separate section in this paper but is used as a baseline in the experiment. I think section 2.3 can be removed and filled by other details of the proposed method.
- Utilization of other datasets? The authors mention several public neuroimaging datasets in the introduction but the proposed framework is validated only on a single benchmark. I believe validating the method on other neuroimaging studies will strengthen the paper.
- It is quite straight forward that task-wise supervision during decomposition will, of course, increase the accuracy of the downstream task prediction. Are there other benefits? What if there is a label set difference between the training and testing set?
- Including supervision in dimension reduction / decomposition has a long history. I think the very very basic baseline should be LDA rather than NMF or SVD as it is a supervised method.
- Perhaps the authors should discuss why SupNMF is outperforming SupSVD in Fig 2.

**Limitations:**

The paper does not discuss any limitation of its own.

---

> ### Author Rebuttal · Authors · 2023-08-09
>
> We thank the reviewer for detailed review and helpful comments.
>
> Weaknesses:
> The review cites weakness in motivation, clarity in presentation, use of only one study, and need for additional baselines. We have corrected all issues relating to presentation and motivation. Furthermore, we have added a new dataset from the Cambridge Center for Aging and Neurosciences (CamCAN). We have also added an additional performance baseline using ICA, which is the most popular technique in neurosciences. These new results demonstrate that our performance gains generalize across datasets, and that they significantly outperform baselines (ICA and prior baselines of NMF and Supervised SVD).
>
> We now address the specific questions raised by the reviewer in the following.
>
> **Q1**:I am not sure what the role of supervised SVD is in this paper. It is written as a separate section in this paper but is used as a baseline in the experiment. I think section 2.3 can be removed and filled by other details of the proposed method.
>
> **A1**: Supervised SVD presents the state-of-the-art baseline in supervised dimensionality reduction. This is the reason for inclusion of Supervised SVD. In response to the reviewer, we have added another baseline (ICA). If the reviewer still recommends removing Section 2.3 and expanding discussion of our method, experimental protocols, and conclusions, we are happy to do so.
>
> **Q2**:Utilization of other datasets? The authors mention several public neuroimaging datasets in the introduction but the proposed framework is validated only on a single benchmark. I believe validating the method on other neuroimaging studies will strengthen the paper.
>
> **A2**: We have now implemented our method on the CamCAN dataset as well, segmented using the Harvard Oxford Atlas (HOA). Results from our method consistently exhibit consistently high task differentiation accuracy and interpretability.
>
> Table 1: Test accuracy on CamCAN dataset (contain 3 tasks) using HOA atlas
>
> ||SupNMF| | |NMF| ||
> |-|-|-|-|-|-|-|
> || KNN | MLP| SVM|KNN | MLP| SVM|
> |rank=6|73.56±4.73|74.04±5.83|73.35±5.35|71.04±6.92|73.22±6.69|72.82±5.42|
> |rank=5|72.77±4.72|73.76±5.45|73.76±5.58|70.96±7.58|73.09±6.96|72.2±5.91|
> |rank=4|72.07±4.68|73.03±4.83|75.59±5.38|69.52±8.4|71.22±7.54|70.53±6.23|
> |rank=3|71.06±3.83|71.6±4.39|75.43±4.52|66.38±9.74|68.4±7.77|67.45±5.61|
>
> **Q3**:It is quite straight forward that task-wise supervision during decomposition will, of course, increase the accuracy of the downstream task prediction. Are there other benefits? What if there is a label set difference between the training and testing set?
>
> **A3**: Our motivation for integrating supervision is not merely task prediction, but rather to uncover canonical patterns in the functional physiology of the human brain while performing tasks. We accomplish this by deconvolving observed connectomic signals into a set of primitive connectomes that are largely unique to tasks. This is the main contribution of our work. The fact that our methods also yield excellent classification accuracy is an added benefit.
>
> We observe that our method finds patterns that are supported by neuroscience experiments reported in literature. For example, regions in the left prefrontal cortex are associated with word and sentence comprehension [16], which is over-represented in A4, only contributing to Language task. Regions in the left prefrontal cortex are associated with word and sentence comprehension [16], which is over-represented in A4 of Fig 6  corresponding to the language task, as shown in S4 of 5. The dorsal Default Mode Network (dDMN) is known to be active during Rest [5]. The anatomical regions for this functional network in the posterior  cingulate cortex (the limbic node), and the angular gyrus found in the posterior part of the inferior parietal lobe, are over-represented in A9 of Fig 6. Additionally, substructures corresponding to the dorsal medial prefrontal cortex are also found in A9. We see that `rest’ connectomes are strongly activated for the corresponding column in the S matrix, as shown in Figure 4. The regions implicated in social processing are the medial prefrontal cortex, which is located in the prefrontal cortex of the frontal lobe [15]. In our results, these nodes are over-represented in A18. Finally, the regions implicated in relational processing are dorsolateral prefrontal cortex, rostrolateral prefrontal cortex, and posterior parietal cortex [23]. These regions are over-represented in A20, and A3 respectively.
>
> In contrast to existing methods, our model offers valuable insights into cognitive tasks, providing both high interpretability and efficiency.
>
> **Q4**:Including supervision in dimension reduction / decomposition has a long history. I think the very very basic baseline should be LDA rather than NMF or SVD as it is a supervised method.
>
> **A4**:While LDA and our model both employ supervision, their objectives differ. LDA seeks linear discriminants to enhance class separation. Although effective for class discrimination, LDA does not yield an interpretable "basis" matrix, a necessary feature of our framework. Our methodology enables the extraction of distinct "building blocks" from the deconvolved data matrix, facilitating subsequent brain region correlation analysis and visualization of associated connectomes. This aids in discerning region-specific functions tied to distinct tasks. A more relevant baseline from the neurosciences community is ICA. We present new results demonstrating the superiority of our method over ICA, which can be found in the author rebuttal PDF.
>
> **Q5**:Perhaps the authors should discuss why SupNMF is outperforming SupSVD in Fig 2.
>
> **A5**: SupSVD's performance is influenced by its orthonormal basis. We observe that this constraint can be limiting, as significant patterns or factors in the data may not necessarily be normal. This explains SupNMF's superior performance in Fig 2. We are happy to include this explanation in the paper.

---

> > ### Comment · Reviewer_zaHM · 2023-08-16
> >
> > Thanks for the thorough rebuttal.
> >
> > My main concern is mainly with baselines with supervision, and ICA does not address this issue. Supervised SVD seems like a quite outdated baseline, and simply searching for supervised non-negative matrix factorization already yields so many literature (not mentioned at all in the related work nor in the introduction) that use supervision or semi-supervision for NMF, so I am not quite convinced where the novelty of the proposed method is coming from. Moreover, including supervised SVD as a separate section is out of scope unless it is a cornerstone of the proposed method.

---

> > > ### Author Response · Authors · 2023-08-16
> > >
> > > Thank you for taking the time to review our submission and providing your insights.
> > >
> > > First and foremost, we are more than willing to include and compare our method to any specific baseline that you deem pertinent. It would be immensely helpful if you could specify which particular baseline you'd like to see compared.
> > >
> > > We'd also like to remind and emphasize that NeurIPS,  has topics of interest that specifically call out neuroscience and cognitive science. Our contributions primarily target the advancement of functional connectomics, which is a significant subfield in neuroscience. This broader perspective might explain why some baselines that seem more mainstream in other domains are not as emphasized in our work. We genuinely believe our research adds value to this niche area.

---

### Official Review · Reviewer_vACv · 2023-07-05

**Soundness:** 3 good
**Presentation:** 3 good
**Contribution:** 3 good
**Rating:** 5
**Confidence:** 2

**Summary:**

The paper presents a new approach to identify task-specific building blocks of neuronal activity from fMRI data by using supervised matrix factorisation. The identified patterns generalise from the train to a test set and match expectations on the brain activity for the different tasks from the neuroscience literature.

**Strengths:**

The paper is well written and addresses an important problem in the analysis of fMRI data in a novel way that achieves impressive performance.

**Weaknesses:**

While the paper criticises that existing methods cannot be applied to large, diverse datasets, the paper lacks a study of the computational efficiency of the proposed approach and a comparison with existing methods.

**Questions:**

- How well does the method scale?
- What results do existing methods achieve in the performed experiments? e.g. ICA-based [13, 10, 34] or other ML-based methods [19, 26, 33, 29]

**Limitations:**

Limitations of the approach are not openly discussed.

---

> ### Author Rebuttal · Authors · 2023-08-09
>
> We thank the reviewer for the detailed review and helpful comments.
>
> Weaknesses: The review identifies weaknesses in the study of computational efficiency and comparison with existing methods. To address these concerns, we have added experiments on a new larger dataset from the Cambridge Center for Aging and Neurosciences (CamCAN) and also added comparisons with ICA, the standard technique used in the neurosciences community (as also suggested by the reviewer). Our results conclusively show the superiority of our methods, as well as scalability to the larger CamCAN dataset. Detailed tables and figures can be found in the author rebuttal PDF.
>
> We now address the specific questions raised by the reviewer in the following.
>
> **Q1** :How well does the method scale?
>
> In the paper, we only provide a sample size of 100 subjects, which is the largest number of "unrelated" subjects in the HCP datasets (in such experiments in neuroscience, it is important to select unrelated subjects to eliminate potential bias). We have performed additional experiments on the larger dataset from the Cambridge Center for Aging and Neuroscience (CamCAN), which shows that our method scales very well to increasing cohort sizes.
>
> We also present scaling results in terms of the number of factors (ranks) from 7 (selected because we have 7 tasks) to 20 (at which point the factors have converged). We present results on the computational scaling in terms of the number of factors to demonstrate that our method scales very well in terms of the number of factors. In general, we have not observed our computational methods to be the scaling bottleneck – the preprocessing pipeline for denoising, motion correction, and alignment takes much longer than our deconvolution methods.
>
> Table 2: Test accuracy using different sample size and rank on HCP dataset
>
> |  rank=20 |   SupNMF   |            |            |     NMF    |            |           |
> |:--------:|:----------:|:----------:|:----------:|:----------:|:----------:|:---------:|
> | subjects |     20     |     50     |     100    |     20     |     50     |    100    |
> |    KNN   | 83.79±2.41 | 90.97±1.26 |  88.7±0.84 | 76.07±9.85 | 76.29±3.79 | 83.5±1.85 |
> |    MLP   | 86.96±5.68 | 89.36±3.11 | 88.39±2.85 | 81.07±5.77 | 86.43±2.58 |  88±2.34  |
> |    SVM   |   88±3.26  | 89.18±2.37 | 88.79±2.17 | 66.43±6.43 | 84.14±2.51 |   87±2.8  |
>
> | rank=15  | SupNMF     |            |            | NMF        |            |            |
> |----------|------------|------------|------------|------------|------------|------------|
> | subjects | 20         | 50         | 100        | 20         | 50         | 100        |
> | KNN      | 81.5±2.78  | 88.24±0.99 | 88.79±0.67 | 56.43±9.29 | 82.43±4.65 | 82.43±3.5  |
> | MLP      | 80.54±7.27 | 84.43±4.93 | 87.29±3.4  | 78.57±8.45 | 83.71±4.87 | 87.14±2.26 |
> | SVM      | 88.21±7.67 | 88.29±3.31 | 89.5±1.24  | 73.93±7.33 | 76.29±6    | 86±2.62    |
>
> | rank=10  | SupNMF      |            |            | NMF         |            |            |
> |----------|-------------|------------|------------|-------------|------------|------------|
> | subjects | 20          | 50         | 100        | 20          | 50         | 100        |
> | KNN      | 83..71±1.91 | 85.72±1.44 | 88.54±0.49 | 70.71±4.74  | 75.14±4.05 | 82.64±2.02 |
> | MLP      | 81.43±7.46  | 88.5±3.92  | 88.14±2.16 | 72.5±8.3    | 83.43±4    | 87.36±2.33 |
> | SVM      | 84.64±7.67  | 88.5±2.83  | 87.64±2.16 | 71.79±11.46 | 76.43±2.49 | 86.86±1.7  |
>
> | rank = 7 | SupNMF     |            |            | NMF        |            |            |
> |----------|------------|------------|------------|------------|------------|------------|
> | subjects | 20         | 50         | 100        | 20         | 50         | 100        |
> | KNN      | 74.33±2.65 | 83.27±2.66 | 85.72±2.07 | 67.14±9    | 78.29±4.77 | 83.29±1.6  |
> | MLP      | 76.07±6.5  | 86.5±5.61  | 86.32±6.47 | 53.21±6.28 | 82.43±5.31 | 87.71±2.19 |
> | SVM      | 83.57±7.35 | 86.29±8.01 | 86.79±6.2  | 63.57±7.28 | 80.14±3.53 | 84.93±2.87 |
>
> **Q2**: What results do existing methods achieve in the performed experiments? e.g. ICA-based [13, 10, 34] or other ML-based methods [19, 26, 33, 29]
>
> As noted in the related work section, our goals are quite different from ICA and ML methods. The other methods referenced above (and in our work) primarily focus on task identification accuracy. In contrast, our goal is to identify the functional basis of tasks in terms of explainable, physiologically grounded connectomes. The high classification accuracy is an added benefit of our work.
>
> In either case, for completeness, we also include a comparison of our method with ICA in this rebuttal (and will add these results to the paper). These results show that our method is consistently more accurate than ICA across a range of cohort sizes and number of factors, while at the same time yielding interpretable results! We thank the reviewer for this comment, which motivated us to further demonstrate the power of our method.
>
> |         |          |     ICA     |             |            |
> |:-------:|:--------:|:-----------:|:-----------:|:----------:|
> |         | subjects |      20     |      50     |     100    |
> |         |    KNN   |  18.93±3.47 |  18.09±1.47 |  15.3±0.94 |
> | rank=20 |    MLP   |  42.5±10.28 | 43.71±12.59 | 39.21±9.96 |
> |         |    SVM   |  24.64±7.04 |  18.43±5.17 | 17.64±2.35 |
> |         |    KNN   |  25.17±7.92 |  11.61±1.23 | 17.34±1.94 |
> | rank=10 |    MLP   | 32.86±10.69 |  20.14±8.37 | 34.93±6.44 |
> |         |    SVM   |  18.57±8.86 |  18.43±4.06 | 16.43±8.97 |

---

### Official Review · Reviewer_Rosr · 2023-07-07

**Soundness:** 4 excellent
**Presentation:** 4 excellent
**Contribution:** 4 excellent
**Rating:** 7
**Confidence:** 4

**Summary:**

This contribution presents a novel method to find a functional basis for a database of task fMRI acquired from different subjects. The functional basis, dubbed canonical task connectomes, is shard across large cohorts; can be composed into task-specific networks; and is predictive of task efficacy.

The authors produce this functional basis through supervised and non-supervised NMF and SVD. For this they propose an objective function (in equation 3) which is compatible with these methodologies.

To implement this approach the authors use the HCP100 database which has 6 cognitive tasks. To show that the obtained basis is task-specific the authors use UMAP plots of different resulting decompositions showing good separability of tasks in the embedded UMAP space. To show that their basis is generalizable across cohorts they use 80/20 splits of the HCP 100 database and use the basis to inform a classifier that predicts the task performed by an unseen subject given the fMRI acquisition. To claim physiological and anatomical grounding for their basis, the authors compare qualitatively the basis against known anatomical traits and the involvement of different basis components as important features for each task.

**Strengths:**

This contribution is a high-quality application of known methods to the significant problem of understanding how functional connectivity in the human brain (as measured  by fMRI) is centric to cognitive tasks.

The manuscript presents a well-justified formulation of the problem as a deconvolution case and solves it through different approaches, supervised and non-supervised. This formulation and resolution are original and well presented. Even if the methodological contribution is not at the center of this manuscript, the application of known techniques is well-justified and evaluated.

The evaluation of the results are a good balance of qualitative evaluation (e.g. with UMAP embeddings), and quantitative (e.g. with the clustering approaches) in the case of task-specificity of the connectomes. The generalisation experiment using a downstream classification task is also well conceived. Finally the relation with anatomy and physiology is well organised.

In all, this contribution presents a very good application of known methods to an important problem in neuroimaging. So it's a paper that will have impact in one area, the neuroimaging one.

**Weaknesses:**

I find two weaknesses which are related to claims of cohort generalisability. In short, with the availability of public datasets of fMRI, it's hard to justify a cohort generalisation claim while staying in one 100-subject database. Specifically when the used 100-subject set is a subsample of a 1,200 total database. In light of this, second weakness I find is the lack of an analysis of the stability of the functional basis across datasets, including a study on dataset size.

**Questions:**

My main questions will be database specific.

First, how many subjects are needed to obtain the functional basis, or canonical task connectomes? For this the authors could provide a learning curve-style analysis analysing the dispersion of the found connectomes with respect to the sample size.

Second, to properly claim cross-cohort the authors should properly use a second large cohort, such as UK Biobank or ABCD which, admittedly, have different task fMRI protocols. Short of this, the authors might look into toning done the cross-cohort claim.

**Limitations:**

The authors have not explicitly mentioned the limitations in the manuscript.

---

> ### Author Rebuttal · Authors · 2023-08-09
>
> We thank the reviewer for the detailed review and helpful comments.
>
> Weaknesses: The review identifies two weaknesses, lack of generalizability across studies and stability of functional basis across cohorts. We have added a new dataset from the Cambridge Center for Aging and Neurosciences (CamCAN) using a different atlas (Harvard-Oxford Atlas), and show that our results generalize across these rather diverse cohorts and that the factors identified by our deconvolution method are stable across studies. Please see the new results included in the PDF of the author's response.
>
> We now address the specific questions raised by the reviewer.
>
> **Q1**: First, how many subjects are needed to obtain the functional basis, or canonical task connectomes? For this the authors could provide a learning curve-style analysis analyzing the dispersion of the found connectomes with respect to the sample size.
>
> **A1**:  In our study, we present results for rank=10 and rank=20 with a sample size of 100. Presented below are additional experimental outcomes for varying ranks and sample sizes:
> As shown in the table, SupNMF yields consistently high performance across varied subject numbers and ranks (number of factors). In contrast, NMF experiences a significant drop in accuracy with fewer subjects.
>
> Table 1: Test accuracy using different sample size and rank on HCP dataset
>
> |  rank=20 |   SupNMF   |            |            |     NMF    |            |           |
> |:--------:|:----------:|:----------:|:----------:|:----------:|:----------:|:---------:|
> | subjects |     20     |     50     |     100    |     20     |     50     |    100    |
> |    KNN   | 83.79±2.41 | 90.97±1.26 |  88.7±0.84 | 76.07±9.85 | 76.29±3.79 | 83.5±1.85 |
> |    MLP   | 86.96±5.68 | 89.36±3.11 | 88.39±2.85 | 81.07±5.77 | 86.43±2.58 |  88±2.34  |
> |    SVM   |   88±3.26  | 89.18±2.37 | 88.79±2.17 | 66.43±6.43 | 84.14±2.51 |   87±2.8  |
>
> | rank=15  | SupNMF     |            |            | NMF        |            |            |
> |----------|------------|------------|------------|------------|------------|------------|
> | subjects | 20         | 50         | 100        | 20         | 50         | 100        |
> | KNN      | 81.5±2.78  | 88.24±0.99 | 88.79±0.67 | 56.43±9.29 | 82.43±4.65 | 82.43±3.5  |
> | MLP      | 80.54±7.27 | 84.43±4.93 | 87.29±3.4  | 78.57±8.45 | 83.71±4.87 | 87.14±2.26 |
> | SVM      | 88.21±7.67 | 88.29±3.31 | 89.5±1.24  | 73.93±7.33 | 76.29±6    | 86±2.62    |
>
> | rank=10  | SupNMF      |            |            | NMF         |            |            |
> |----------|-------------|------------|------------|-------------|------------|------------|
> | subjects | 20          | 50         | 100        | 20          | 50         | 100        |
> | KNN      | 83..71±1.91 | 85.72±1.44 | 88.54±0.49 | 70.71±4.74  | 75.14±4.05 | 82.64±2.02 |
> | MLP      | 81.43±7.46  | 88.5±3.92  | 88.14±2.16 | 72.5±8.3    | 83.43±4    | 87.36±2.33 |
> | SVM      | 84.64±7.67  | 88.5±2.83  | 87.64±2.16 | 71.79±11.46 | 76.43±2.49 | 86.86±1.7  |
>
> | rank = 7 | SupNMF     |            |            | NMF        |            |            |
> |----------|------------|------------|------------|------------|------------|------------|
> | subjects | 20         | 50         | 100        | 20         | 50         | 100        |
> | KNN      | 74.33±2.65 | 83.27±2.66 | 85.72±2.07 | 67.14±9    | 78.29±4.77 | 83.29±1.6  |
> | MLP      | 76.07±6.5  | 86.5±5.61  | 86.32±6.47 | 53.21±6.28 | 82.43±5.31 | 87.71±2.19 |
> | SVM      | 83.57±7.35 | 86.29±8.01 | 86.79±6.2  | 63.57±7.28 | 80.14±3.53 | 84.93±2.87 |
>
> Accompanying visualizations illustrating task accuracy spanning ranks 7 through 20 and varying subject counts are provided in the author rebuttal PDF. This enhanced analysis will be incorporated into the final version.
>
> **Q2**: Second, to properly claim cross-cohort the authors should properly use a second large cohort, such as UK Biobank or ABCD which, admittedly, have different task fMRI protocols. Short of this, the authors might look into toning down the cross-cohort claim.
>
> **A2**: Addressing the cross-cohort question, we implemented our methodology on the CamCAN dataset, segmented using the Harvard Oxford Atlas (HOA). The outcomes consistently exhibit reliable task differentiation and interpretability, supporting the generalizability claims in the paper. And you can find the figure showing clear discrimination on CamCAN tasks in the author rebuttal PDF.
>
> Table 2: Test accuracy on CamCAN dataset (contain 3 tasks) using HOA atlas
>
> |           |SupNMF         |                 |                       |   NMF         |                    |                    |
> |---------|-------------------|---------------|-------------------|----------------|-----------------|------------------|
> |           | KNN             | MLP            | SVM             |    KNN        | MLP            | SVM            |
> |rank=6| 73.56±4.73  | 74.04±5.83 | 73.35±5.35    | 71.04±6.92 | 73.22±6.69 | 72.82±5.42 |
> |rank=5| 72.77±4.72  | 73.76±5.45 | 73.76±5.58    | 70.96±7.58 | 73.09±6.96 | 72.2±5.91  |
> |rank=4| 72.07±4.68  | 73.03±4.83 | 75.59±5.38    | 69.52±8.4  | 71.22±7.54 | 70.53±6.23 |
> |rank=3| 71.06±3.83  | 71.6±4.39   | 75.43±4.52     | 66.38±9.74 | 68.4±7.77  | 67.45±5.61 |

---

> > ### Comment · Reviewer_Rosr · 2023-08-19
> >
> > Thanks to the authors for the responses to my questions. I will now update my score to accept.

---

### Official Review · Reviewer_jVGg · 2023-07-07

**Soundness:** 2 fair
**Presentation:** 3 good
**Contribution:** 3 good
**Rating:** 4
**Confidence:** 4

**Summary:**

This paper presents a novel framework for fMRI analysis that aims to deconvolve complex neuronal networks into task-specific elementary networks called "canonical task connectomes." The proposed method utilizes supervised matrix factorization to identify these task-specific networks and demonstrates their interpretability and generalizability. The study showcases experimental results on the Human Connectome Project dataset, highlighting the ability of the framework to capture the natural task-specific structure in neuroimages.

**Strengths:**

- The paper presents a new problem formulation and introduces the SupNMF method, which is a novel approach to identifying task-specific networks. The authors demonstrate the usefulness of the proposed framework in identifying canonical task connectomes that have a strong physiological basis and can be mapped to regions of the brain to identify physiological underpinnings of tasks.
- the authors present the problem formulation and the proposed method in a clear and concise manner.
- The proposed interpretable framework has the potential to advance understanding of complex cognitive processes and to identify biomarkers for predicting tasks.
- The authors also provide a comprehensive discussion of relevant methods and materials.

**Weaknesses:**

- While the authors present comprehensive experimental results, they could provide more details on the performance of the proposed framework in comparison to other state-of-the-art methods. Additionally, the authors could provide more details on the interpretability of the identified canonical task connectomes and how they relate to existing literature in neurosciences.
- While the authors briefly mention the potential applications of the framework in understanding shared and unique functional networks across different pathologies and how task-specific networks can get dysregulated due to the onset and progression of diseases, a more in-depth discussion of these applications and their potential impact on the field would be helpful.
- The authors did not explain much on why the “unrelated set” of subjects in the Human Connectome Project is selected. Also, more datasets are expected to be included to demonstrate the generalizability of the proposed method.
- The authors could provide more details on how they determined the optimal number of latent connectomes and how this choice impacts the results. One potential concern of the proposed framework is that it relies on the assumption that the observed connectome matrix can be represented as a linear combination of a small number of latent matrices. While this assumption may hold for some datasets, it may not be applicable to all fMRI datasets, especially those with high levels of noise or variability. Additionally, the choice of the number of latent connectomes (i.e., the dimensionality of latent space) is critical and may impact the performance of the proposed framework.
Another potential weakness of the proposed framework is that it requires task-label vectors for each connectome in the dataset. While the authors provide details on how they obtained the task-label vectors for the HCP dataset, it may not be feasible to obtain such labels for all fMRI datasets. Additionally, the choice of the task-label vectors may impact the performance of the proposed framework, and the authors could provide more details on how they selected the task-label vectors and how this choice impacts the results.

**Questions:**

- Could you provide a more detailed comparison of the performance of the proposed framework with more state-of-the-art methods of other types (CNNs, GNNs)? How does the proposed framework outperform or differ from existing approaches in terms of accuracy and generalizability?
-  Can you elaborate more on the interpretability of the identified canonical task connectomes? How do these connectomes relate to existing literature in neurosciences? Are there any specific brain regions or networks that are consistently identified across different tasks?
- How generalizable are the findings of this study? Are there any potential biases or confounding factors that could impact the results?
- How does the proposed framework handle potential confounding factors such as motion artifacts or physiological noise? Were any specific preprocessing steps or techniques employed to address these confounds?



**Limitations:**

While the proposed framework has the potential to advance our understanding of complex cognitive processes and to identify biomarkers for predicting tasks, it is important to consider the potential ethical implications of this research. For example, the use of fMRI data for predicting cognitive states or identifying biomarkers could raise concerns about privacy, informed consent, and potential misuse of the data.

---

> ### Author Rebuttal · Authors · 2023-08-09
>
> We thank the reviewer for detailed review and helpful comments.
>
> Weaknesses:
>
> 1.Comparison to state of the art methods: We have also added comprehensive experiments and comparisons to ICA, which is the most commonly used method in this domain. Please see the results in author rebuttal PDF. Our superior results firmly establish the benefits of our method compared to the current best/commonly used methods in the area.
>
> 2.Interpretability: We have now added a section on interpreting the factors and their mapping to human brain regions and a discussion of how these regions are known to function for corresponding tasks.
>
> 3.Linearity assumptions: This is indeed a valid consideration. However, we submit that ours is the first such investigation on deconvolving brain connectomes into interpretable factors in the area and our results are highly promising notwithstanding our linearity assumption. Indeed, we expect follow-on efforts that may investigate non-linear superpositions as well.
>
> 4.Unrelated subjects: This is indeed the norm in the area to eliminate any biases arising from physiological similarities among related subjects.
>
> 5.Generalizability to other datasets: We have added experiments on a new dataset – the Cambridge Center for Ageing and Neuroscience (CamCAN) along with a new atlas – the Harvard-Oxford Atlas. Our results show excellent generalizability beyond the Human Connectome Project dataset in our original submission.
>
> 6.Choice of the number of factors: we now show that our results are robust across choices of factors (please see the results in PDF).
>
> We now address the specific questions raised by the reviewer in the following.
>
> **A1**: Our framework is fundamentally different from CNNs or GNNs, which focus on prediction accuracy. However, these methods cannot identify specific sub-connectomes, associated anatomical parts of the brain, and their contribution to various tasks. The latter is a key motivation for our proposed work. We discuss this in our response to Q2 in detail.
>
> **A2**: This is an important question – one that provides motivation, and is the main contribution of our work. As a concrete example of interpretability, after we construct the “basis matrix” A  of size (64620,20) by deconvolving data matrix X, we have the 20 "building blocks" A1-A20 (20 columns in A) for different tasks. The number of rows, 64620, is the dimension that corresponds to the region index, which allows us to map each of the 20 factors back to the brain and obtain a $360 \times 360$ region correlation matrix. We then apply the BioImageSuite tool on these correlation matrices using node definitions from the atlas of Glasser et al. 2016 to visualize each of the 20 connectomes. By examining the connectomes (i.e., each of the 20 factors), we uncover the region functions associated with specific tasks.
>
> Examining the factors (A1-A20), we find that there is a very clean separation for different tasks (in other words, factors are strongly associated with tasks). In Fig. 5, we can see all factors contributing strongly to only 1-2 tasks. This suggests two important facts: (i) the factors correspond to the functional physiology, as it relates to the tasks; and (ii) localization of the functional regions can be used for task discrimination using inexpensive modalities, such as EEGs. These physiological associations are partially confirmed by domain studies – for example, regions in the left prefrontal cortex are associated with word and sentence comprehension [16], which is over-represented in A4, only contributing to Language task. Regions in the left prefrontal cortex are associated with word and sentence comprehension [16], which is over-represented in A4 of Fig 6  corresponding to the language task, as shown in S4 of 5.
>
> For the last question “Are there any specific brain regions or networks that are consistently identified across different tasks?”. Since we examine the problem of task-specific factors, we aim to find regions that discriminate across different tasks and remove the noise and the basal signal in our brains. Indeed, our method can be easily modified to extract these background signals as well.
>
> In summary, compared with existing methods, our framework provides interpretable and discriminating signals that provide the functional basis of various tasks. We attach figures and additional explanations in our PDF rebuttal.
>
> **A3**: Scale and generalizability are indeed important issues. Here, we provide more experimental details on these aspects:
>
> For generalization, we add additional results on the robustness of our method by examining our results across the full range of number of factors from 7 to 20 (7 because we have 7 tasks in HCP and 20 because the dominant factors saturate by rank 20). Our results are remarkably robust across this range of number of factors. Furthermore, with respect to generalizability beyond datasets, we have now added another dataset CamCAN, which confirms the superiority of our methods across two vastly different cohorts, acquisition protocols, MRI platforms, and atlases. Please see all detailed results in PDF.
>
> Our experimental protocols (use of unrelated subjects), unbiased sampling of subjects, and use of different acquisition protocols, atlases, and instrumentation, aim to eliminate other confounders.
>
> **A4**: As expected, fMRI data from HCP has significant noise, including motion artifacts and physiological variability, which is the reason our problem is complex. We use a sophisticated pipeline for denoising, motion correction, and registration to cancel noise, head motion, and instrumentation effects. We then register all of the images to a reference model (MNI) so we can apply an atlas for segmenting the brain to different regions. Our excellent results indicate that these pipelines are able to significantly reduce noise and variability.

---

> ### Comment · Reviewer_jVGg · 2023-08-21
> **Response to Rebuttal**
>
> Thanks the authors for the reply. I believe this work presents some interesting insights on interpretable connection analysis.

---

### Author Rebuttal · Authors · 2023-08-09

We thank the reviewers for their positive and constructive feedback.
Now we address some common questions raised by the reviewers in the following two aspects:
1. Motivation and contribution of this work
Discriminating tasks serve as a downstream objective following the identification of physiologically interpretable cognitive "building blocks" within task connectomes. Several inquiries have arisen regarding task accuracy and its comparison to other classification techniques, including CNN/GNN, PLS, and LDA. We aim to clarify our primary motivation and contribution: to discern the fundamental neural patterns enabling diverse complex cognitive tasks. Our methodology identifies patterns corroborated by existing neuroscience research. For instance, regions within the left prefrontal cortex, prominent in A4, are uniquely linked to language tasks, underscoring their association with word and sentence comprehension [16]. Compared with existing methods, our framework provides meaningful insight to uncover the functional basis of various tasks, and in the process, also yields excellent task discrimination.


2. Generalization and scale
To address concerns of generalization and scalability, we evaluated variations in both subject sizes and factor numbers (rank) and expanded our tests to include the CAMCAN dataset and different atlases. Comprehensive tables and figures are available in the attached PDF.
The table indicates that SupNMF consistently maintains robust performance irrespective of changes in subject count or rank (factor numbers). Conversely, NMF's accuracy noticeably declines with a reduced number of subjects. We have also provided visual illustrations showcasing task accuracy across ranks from 7 to 20 factors and different subject sizes. This extended analysis will be integrated into the paper's final edition.
In our study, we employed 100 "unrelated" subjects from the HCP dataset, aligning with the standard practice in the domain which is the maximum count obtainable from the HCP dataset. This selection was made to preclude any bias from physiological resemblances in related subjects.

---

### Decision · Program_Chairs · 2023-09-21

**Decision:**

Reject

**Comment:**

The paper introduces SupNMF method, a novel approach to identifying task-specific networks; such networks are expected to a strong physiological basis. The problem is well-formulated and the method is presented in a clear and concise manner. Conceptually, it helps understanding how functional connectivity in the human brain (as measured by fMRI) is centric to cognitive tasks.

Weaknesses related to claims of cohort generalisability have been outlined. With the availability of public datasets of fMRI, experiments on  one 100-subject database is too limited. Comparison with state of the art methods is lacking. In particular the authors seem to have missed important related work based on sparse PCA/dictionary learning. In view of the state of the art, the methodological contributions in the paper seem modest. There also remain important issue regarding model selection.

The best thing to do is to encourage the authors to resubmit it to another conference with some improvements included.